# Alteration of Biomolecular Conformation by Aluminum-Implications for Protein Misfolding Disease

**DOI:** 10.3390/molecules27165123

**Published:** 2022-08-11

**Authors:** Yuhai Zhao, Aileen I. Pogue, Peter N. Alexandrov, Leslie G. Butler, Wenhong Li, Vivian R. Jaber, Walter J. Lukiw

**Affiliations:** 1LSU Neuroscience Center, Louisiana State University Health Science Center, New Orleans, LA 70112, USA; 2Department of Cell Biology & Anatomy, LSU Health Science Center, New Orleans, LA 70112, USA; 3Alchem Biotek Research, Toronto, ON M5S 1A8, Canada; 4Russian Academy of Medical Sciences, 113152 Moscow, Russian; 5Department of Chemistry, Louisiana State University, Baton Rouge, LA 70803, USA; 6Department of Pharmacology, Jiangxi University of TCM, Nanchang 330004, China; 7Department of Ophthalmology, LSU Health Science Center, New Orleans, LA 70112, USA; 8Department Neurology, LSU Health Science Center, New Orleans, LA 70112, USA

**Keywords:** aluminum, Alzheimer’s disease (AD), biomolecules, adenosine triphosphate (ATP), histone linker proteins (H1 class), prion disease (PrD), protein folding disease

## Abstract

The natural element aluminum possesses a number of unique biochemical and biophysical properties that make this highly neurotoxic species deleterious towards the structural integrity, conformation, reactivity and stability of several important biomolecules. These include aluminum’s **(i)** small ionic size and highly electrophilic nature, having the highest charge density of any metallic cation with a Z^2^/r of 18 (ionic charge +3, radius 0.5 nm); **(ii)** inclination to form extremely stable electrostatic bonds with a tendency towards covalency; **(iii)** ability to interact irreversibly and/or significantly slow down the exchange-rates of complex aluminum–biomolecular interactions; **(iv)** extremely dense electropositive charge with one of the highest known affinities for oxygen-donor ligands such as phosphate; **(v)** presence as the most abundant metal in the Earth’s biosphere and general bioavailability in drinking water, food, medicines, consumer products, groundwater and atmospheric dust; and **(vi)** abundance as one of the most commonly encountered intracellular and extracellular metallotoxins. Despite aluminum’s prevalence and abundance in the biosphere it is remarkably well-tolerated by all plant and animal species; no organism is known to utilize aluminum metabolically; however, a biological role for aluminum has been assigned in the compaction of chromatin. In this Communication, several examples are given where aluminum has been shown to irreversibly perturb and/or stabilize the natural conformation of biomolecules known to be important in energy metabolism, gene expression, cellular homeostasis and pathological signaling in neurological disease. Several neurodegenerative disorders that include the tauopathies, Alzheimer’s disease and multiple prion disorders involve the altered conformation of naturally occurring cellular proteins. Based on the data currently available we speculate that one way aluminum contributes to neurological disease is to induce the misfolding of naturally occurring proteins into altered pathological configurations that contribute to the neurodegenerative disease process.

## 1. Introduction

The biosphere-abundant element aluminum [Al^3**+**^; Al(III)] is a pervasive, pro-inflammatory, metallic toxin that is being increasingly mobilized into our environment [1,2,3]. The high positive charge density, small atomic and ionic size, unchanging valence of 3^+^, ubiquity in the biosphere and the extreme affinity of aluminum for electronegative targets and/or oxygen-donor ligands appears to underlie the basis for aluminum’s toxicological properties and the generation of oxidative stress. Pathologically, aluminum: **(i)** is associated with the induction and production of reactive free radicals and reactive oxygen species (ROS) which can overwhelm the antioxidant defenses of the cell and so can both initiate and perpetuate cellular damage; **(ii)** as a native or hydroxylated species can interact directly with cellular components resulting in the generation of functionally stabilized and/or altered or modified biomolecules that impart defective operational capabilities towards cellular homeostasis; **(iii)** appears to interact selectively with cellular components only where the three-dimensional configurations are favorable with compartmentalization into specific regions of the cell and nucleus; and **(iv)** preferentially interacts wherever the coordination chemistry is energetically the most favorable. For example, it has been shown that the extremely high positive charge density aluminum species specifically targets certain accessible, polyphosphate- and adenine + thymine (A+T)-rich, highly electronegative open or ‘euchromatic’ regions of brain cell nuclei to disrupt gene expression patterns [2,3,4,5,6,7]. Importantly, aluminum appears to be compartmentalized into specific molecular, cellular and/or nuclear structures and its concentration may actually be highly localized, and at a higher concentration than those found on average and at random determinations throughout the entire biological or tissue samples under study [8,9,10]. Conversely, very small amounts of aluminum are required to elicit profound neurotoxic effects, ***so the overall determination of aluminum in bulk biological samples may not provide an accurate indicator of aluminum’s potential involvement in any particular disease process***. Many of the cellular, genetic, epigenetic, molecular, nuclear and systemic mechanisms by which aluminum exerts selective neurotoxicity and/or genotoxicity remains incompletely understood [8,9,10,11,12,13,14,15,16,17,18,19,20,21,22,23,24,25,26,27,28,29,30,31,32]. This Communication illustrates three important, interrelated and fairly well characterized molecular and cellular nodes at which aluminum significantly interacts and perturbs normal cellular function. These include: **(i)** preferential interactions with the phosphates and/or polyphosphates of adenosine triphosphate (ATP), that along with other nucleoside triphosphates are the main molecular energy carriers of the cell; **(ii)** the cross-linking of certain linker histone-DNA-associated chromatin domains; and **(iii)** aluminum’s remarkable capability to support altered protein conformations that drive the misfolding of susceptible and potentially pathogenic proteins. This latter property of aluminum may be important for pathological protein aggregation and in “***conformational disease***’, ‘***protein misfolding disease***’ or ‘***protein aggregation disease***’ [32,33,34,35,36,37,38]. These types of interactions appear to be the basis for an expanding list of progressive and lethal human neurodegenerative disorders that include the tauopathies, frontotemporal dementia (FTD), Alzheimer’s disease (AD) and multiple prion disorders (PrD) [34,35,36,37,38,39,40,41].

## 2. Aluminum (Al^3+^) and Adenosine Triphosphate (ATP)

In most cases the mode of aluminum [Al^3+^; Al(III); atomic mass 26.9815; neutral atom electron configuration: 1s^2^2s^2^2p^6^3s^2^3p^1^] interaction with commonly encountered biomolecules remains incompletely understood, but a recurring theme is a strong aluminum interaction with oxygen-donor ligands such as phosphate groups [3,4,5]. Phosphate groups can be available as free species, for example as nucleotide triphosphates such as adenosine triphosphate (ATP), as phosphoproteins involving amino acids with phosphate attached to their side chain (R) groups, such as in cytoskeletal and synaptic proteins, and as a major structural component of nucleic acids including all forms of ribonucleic acid (RNA) and deoxyribonucleic acid (DNA), the genetic material contained within all cell types. Similar in size to the abundant and natural activator magnesium (Mg^2+^), Al^3+^ may act by substituting for Mg^2+^ in vital energy-dependent processes. Virtually all ATP-associated reactions utilize Mg^2+^, and the presence of Al^3+^ potentially irreversibly interferes with these reaction processes and has been shown to limit the availability of ATP in energy-requiring reactions and functions within both the cellular and nuclear compartments (Figure 1) [1,2,3,4,5,6].

## 3. Al^3+^, DNA and Linker Histone H1° 

In aqueous solutions, aluminum [Al^3+^; Al(III)] salts and hydroxides are exceptionally potent aggregators of organic/biological molecules, often coalescing molecular species to the point that they precipitate out of solution. In fact aluminum, as alum [potassium aluminum sulfate; KAS; KAl(SO_4_)_2_·12 H_2_O] is used globally to aggregate organic/biological impurities in turbid drinking water in order to precipitate them and to clarify an unappealing cloudy water product into a clear, finished water product [3,7,11,12,13]. To accomplish this, raw waters are treated with alum that serves as a flocculant; raw water often holds tiny suspended particles that are very difficult to filter and remove, and alum causes them to clump together so that they can settle out of the water and be easily trapped by standard filtration methods [12,13,14]. Interestingly, the speciation of aluminum is often transformed during the processes of coagulation, flocculent formation, filtration and sedimentation and the use of aluminum in the purification of drinking water has long been criticized due to potentially toxic and especially neurotoxic effects on human biology and physiology, and particularly neurobiology and neurophysiology [13,14,15].

At the molecular level aluminum has a remarkable effect on the compaction of DNA and chromatin misfolding into higher order structures, effects that are not observed when natural chromatin is treated with other bioavailable divalent or trivalent metals such as Mg^2+^, Cu^2+^, Ca^2+^, Mn^2+^, Fe^2+^, Fe^3+^ or Cr^3+^ [16,17,18,19]. Chromatin is essentially the assembly of genomic DNA, DNA-binding proteins and other nucleoproteins packaged into the nucleus of eukaryotic cells which together are crucial in regulating a myriad of gene expression programs and essential cellular processes unique to each cell type. It should be kept in mind that the human genome has a very complex organization and the genetic material within chromatin is in a constant dynamic motion, in part because genes are being continuously activated or deactivated with biologically useful divalent metal ions including Ca^2+^, Mg^2+^ and/or Mn^2+^ [20,21,22]. One well-studied aspect of aluminum is its interaction with nucleic acids, specifically RNA, DNA and chromatin and aluminum’s second-to-none capability to irreversibly compact nucleic acid-protein complexes [16,17,18]. The ability of aluminum to alter DNA stability, topology and conformation and compact chromatin ultimately results in quenching the natural actions of the RNA polymerases (RNAPs). These RNAPs normally transcribe DNA and chromatin into messenger RNA (mRNA) and other RNA species that include ribosomal RNA (rRNA), small non-coding RNA (sncRNA), transfer RNA (tRNA) and microRNA (miRNA), all of which are involved in the complex regulation of eukaryotic gene expression [8,9,10,19,23,24,25,26].

Some major studies from our laboratories and those of our colleagues over the last 38 years have focused on human linker histones, sometimes referred to as the H1 linker histone family and their binding to DNA in the presence of aluminum salts [2,3,4,5,6,7,11,12,13,14,15,16,17,18,23,24,25,27,28,29,30]. Histone H1′s and the linker histones are a family of dynamic DNA-binding and chromatin compacting nucleoproteins composed of multiple subspecies each having a unique amino acid sequence essential for higher-order chromatin organization and the regulation of gene transcription. The electrostatic binding of chromatin H1 linker proteins promotes a higher order chromatin compaction and induces a shift from transcribed ‘euchromatin’ into silent ‘heterochromatin’, sometimes referred to as ‘compacted’ or ‘heavy chromatin’. Using electrostatic affinity experiments, high-field 19.6T **^27^**Al solid-state MAS NMR spectroscopy, computer-assisted modelling, bond angle, length and strength algorithms and statistical analysis a model involving the H1 linker family subspecies H1° binding to DNA, our group devised an atomic-molecular model of H1°-DNA binding which may explain the increased compaction of the genetic material as is observed in both experimental aluminum-induced encephalopathy and dialysis dementia as well as in AD-affected brain (Figure 2) [16,17,18,32,33,34]. This H1°-DNA binding model involves the unique adjacent aspartic-98 (Asp98; D98) and glutamic-99 (Glu99; E99) amino acid residues of the ~21.4 kDa, ~194 amino acid human H1° and one of the many accessible phosphate groups of an accessible ‘open’ region of promoter DNA (https://www.ebi.ac.uk/interpro/entry/IntePro/IPR005819/; https://www.genecards.org/cgi-bin/carddisp.pl?gene=H1-0; https://www.degruyter.com/document/doi/10.1515/BC2005.064/html (last accessed on 13 June 2022); Figure 2). The unscheduled and pathogenic binding of specific chromatin proteins such as the linker histone H1° and perhaps other nucleoproteins, and DNA condensing agents such as aluminum to this natural and highly dynamic system appears to shut down the homeostatic expression of the genetic material with an ensuing reduction in the generation of mRNA and other types of RNA as is widely observed in aluminum-treated human neurons in primary cell culture and in multiple neurological disease states [3,10,17,18,20,28].

## 4. Al^3+^, Protein Misfolding and the PrPc to PrPsc Transition in Neurodegenerative Disease

Certain naturally occurring brain- and CNS-abundant peptides and proteins including the microtubule-associated protein tau (MAPT) and the 42 amino acid amyloid beta (Aβ42) peptide have the ability over time and under pathological conditions to assume atypical conformations, thereby altering their normal biological structure and function [34,35,36,37,38,39,40,41]. This causes these brain-enriched species to assume pathological conformations causing them to aggregate into highly insoluble, pro-inflammatory and neurotoxic intracellular lesions, defining the processes known as ‘***conformational disease***’, ‘***protein misfolding disease***’ or ‘***protein aggregation disease***’ [34,35,36,37,38,39,40,41]. Normally, PrPc, a ubiquitously expressed glycosylphosphatidylinositol (GPI)-anchored cell surface glycoprotein, is associated with lipid raft components and functions as a signaling molecule during neuronal development, synaptic plasticity and neuronal myelin sheath maintenance, with additional roles in metal uptake and homeostasis [34,35,36,37,38,39]. The transition of PrPc to PrPsc supports the formation of intracellular lesions whose abundance is linked to progressive inflammatory neurodegeneration, neurological dysfunction, neurobehavioral deficits and disturbances in cognition and progressive dementia [34,35,36,37,38,39,40,41,42,43,44]. These classes of misfolded and aggregated host proteins thereby contribute to the pathogenesis of several progressive, age-related and ultimately lethal human neurodegenerative and dementing amyloidopathies and tauopathies. These encompass a continuous spectrum of brain diseases that include multiple prion disorders in mammals including ‘mad cow disease’ (bovine spongiform encephalopathy; BSE), other human prion diseases that include Creutzfeldt–Jakob disease (CJD), Gerstmann–Sträussler–Scheinker Syndrome (GSS) and fatal familial insomnia (FFI), Parkinson’s disease (PD), triplet-repeat disease (TRD), Alzheimer’s disease (AD), frontotemporal dementia (FTD) and other fatal neurodegenerative diseases known as transmissible spongiform encephalopathies, which affect humans, deer, sheep and cattle [34,35,36,37,38,39,40].

The molecular mechanism of the pathological phenomenon of neuronal protein misfolding lends support to the ***‘prion hypothesis’***, which predicts that the aberrant folding of endogenous natural protein structures into unusual pathogenic isoforms can induce the atypical folding of other similar brain-abundant proteins. This underscores the age-related, progressive nature and potential transmissible and spreading capabilities of the aberrant protein isoforms that drive these invariably fatal neurological syndromes. Divalent metals are known to promote PrPsc formation, and evidence is emerging that a pathological interaction of the environmentally abundant metal ions and/or their oxides with the amino acids of prion proteins are involved in multiple protein misfolding pathways [39,40,41,42,43,44,45,46,47,48,49,50]. Several independent lines of evidence lend support for an effect of environmental metals, metal ions and their oxides on the PrPc to PrPsc transition and come from a number of independent experimental observations: 

**(i)** It is clear that there exist ‘intramolecular regions’ within cellular components with a very high sensitivity and selectivity for aluminum interaction, but only for certain energetically-favorable sites in protein or peptide subtypes/species and genetic environments. Several well-defined areas appear to occur in the oxygen-donor groups of ATP (Figure 1) and in adjacent aspartic-glutamic acid (ASP-GLU) amino acid side groups of certain human H1 linker histone subtypes (Figure 2). Interestingly, human prion protein, enriched in ASN and GLN residues also contains multiple ASP-GLU and GLU-ASP amino acid motifs (https://www.uniprot.org/uniprot/P04156 (last accessed on 13 June 2022). The prion structural conversion from PrPc to PrPsc appears to be driven by the presence of GLN- and ASN-enriched amino acid segments which account for the particle’s priogenicity [45,46,47,48];

**(ii)** Mammalian prion diseases (PrD), especially of deer, sheep and cattle are transmissible via environmental routes and specifically through an environment-mediated transmission via the lithosphere and SiAl, the latter representing the upper layer of the earth’s crust consisting of soils enriched in aluminum silicate [43,44,45,46]. In fact, aluminum-enriched soils are likely to be an important environmental reservoir of prion infectivity and soil-immobilized prions have enhanced infectivity via the oral route compared to unbound prions [45,46,47,48]. The oxide surface of soil particles that include aluminum- and iron-oxides (Al_2_O_3_, Fe_2_O_3_) and various forms of environmentally- and physiologically-available aluminum hydroxides (Al(OH)_3_; see Figure 2) have been implicated in PrP disease transmission affecting prion transport, bioavailability and persistence in soil environments ([44,45,46,47,48,49,50]; unpublished observations). Quartz crystal microbalance with dissipation monitoring analysis and optical waveguide light mode spectroscopy further indicate that based on pH and ionic strength the efficiency of prion attachment to Al_2_O_3_ is in a manner consistent with electrostatic forces dominating PrP interaction with these oxides; and that the presence of the PrP N-terminal promotes strong electrostatic attachment to Al_2_O_3_. This suggests that prions have an affinity for and tendency to associate with Al_2_O_3_ and other charged minerals in soils and/or other metal oxides, again underscoring the attraction of PrP for aluminum and other metal oxide surfaces [46,47,48,49,50];

**(iii)** Tau proteins are misfolded and aggregated in the tauopathies that include AD and FTD; accelerated tau aggregation, apoptosis and neurological dysfunction are induced by a chronic oral administration of aluminum in multiple murine models of tauopathy and induced inflammatory neurodegeneration [51,52,53,54,55,56,57,58,59];

**(iv)** Multiple independent laboratories have studied the effects of supplementing the diets of amyloid-over-expressing transgenic murine models of AD including the Tg2576, APP/PS1, 5xFAD series with aluminum (as chloride, lactate, maltolate or sulfate) and have evaluated the impact of ingested aluminum salts with pathological outcome [5,7,10,26,29,41,42,43,44,47,51,52,53,54,55,56,57,58,59]. A general consensus of the results of these studies is that the presence of aluminum in the diets of TgAD murine models enhances neurodegenerative pathology by progressively intensifying the prevalence of oxidative stress and by increasing the generation of Aβ42 peptides, accelerating Aβ42 oligomerization, amyloidogenesis, aggregated tau and/or senile plaque and NFT deposition [5,7,10,25,26,42,43,44,52,53,54,55,56,57,58,59];

**(v)** Aluminum-fed TgAD models also exhibit an up-regulation in the abundance of pro-inflammatory and pathogenic microRNAs (miRNAs) many of which are involved in amyloidogenesis and neuroinflammation, and are the same miRNA species as those observed to be up-regulated in the brain and CNS of patients suffering from several age-related neurodegenerative disorders [10,18,52,53,54,57]; and

**(vi)** Using Western gel Tris-glycine sodium-dodecyl sulfate (TGSDS)-based electrophoretic analysis, immunoblotting (to distinguish between the PrPc and PrPsc isoforms), immunocytochemistry, immune-fluorescence using PrPsc Antibody (3F4; Alexa Fluor^®^ 700) Novus Biologicals; Centennial, CO, USA; CD230 (PrP) monoclonal antibody (4D5), Invitrogen ThermoFisher Scientific, Waltham, MA, USA and others] and computer-aided algorithms, statistical and imaging analysis, preliminary unpublished evidence from our laboratory suggests that aluminum (as aluminum sulfate; Al_2_(SO_4_)_3_) at nanomolar concentrations strongly promotes a shift in conformation from the PrPc into the PrPsc isoform both in vitro and in human neuronal-glial (HNG) cells in primary co-culture. As such aluminum salts may be an integral component in the ***molecular-metallic*** mechanism involved in inducing protein misfolding disease (see manuscript text; Figure 3). Taken together these findings continue to underscore a potential role for aluminum salts in driving neurotoxicity and neuropathological mechanisms associated with PrPc- to PrPsc-type protein transitions and the initiation and/or propagation of progressive, age-related neurological and neurodegenerative disease states.

## 5. Discussion

There has long been an interest in aluminum as a pathogenic factor in human biology, and especially in neurobiology and neurology, largely because: **(i)** of aluminum’s unusually high positive charge density, unique biochemical and biophysical properties under physiological conditions and within complex biological systems; **(ii)** this metallotoxin’s ubiquity as the most abundant metal in the earth’s biosphere and enrichment in naturally-occurring biological systems; **(iii)** of aluminum’s wide spread bioavailability in drinking water, food, medicines, cosmetics and other consumer products and in atmospheric dust and groundwater that promotes extensive human exposure; **(iv)** multiple investigators have demonstrated aluminum’s extreme neurotoxicity on multiple neurobiological systems, even at very low ambient (nM or lower) levels; and **(v)** of aluminum’s ability to aggregate naturally occurring cellular proteins and peptides, a process known to drive neuro-inflammation, alter neurogenesis and neuronal cytoarchitecture and induce significant memory and learning deficits in multiple experimental murine models [55,56,57,58,59,60,61].

Although aluminum is especially abundant in the biosphere and is remarkably well tolerated by all plant and animal species, no organism is known to use aluminum in any active metabolic process. There is, however, abundant evidence for a biological role for aluminum in the packaging of genes into inactive chromatin conformation [2,3,16,17,32]. Biologically, this is very important because, for example, while all cells of the human body contain the same number and distribution of genes, not all genes are expressed in every individual cell type. For instance, the α-crystallin genes of the eye are not expressed in erythrocytes and the hemoglobin genes of immature erythrocytes are not expressed in the eye, but instead are apparently compacted away into dense, inactive ‘heterochromatic’ configurations by targeted aluminum binding. In fact, aluminum has the highest capability for compacting light transcriptionally active ‘euchromatin’ into inactive ‘heterochromatin’ of any divalent or trivalent metal complex known [16,17,18,19,23,57]. The potential of additional biological roles for aluminum is of continuing interest and studies continue in both human health and disease and in tissue culture and model animal systems. 

## 6. Conclusions

In conclusion, the pathological role for aluminum in neurological disorders has been controversial largely due to an incomplete understanding of aluminum’s precise mode of interaction with select biomolecules and its targeting to specific biological compartments. Many independent researchers have uncovered highly interactive aspects of aluminum neurotoxicity and the precise details of these molecular-metallic pathological mechanisms are becoming increasingly clear. Several seriously understudied areas of aluminum neurotoxicology are: **(i)** the interaction of aluminum with other environmentally-available metals such as mercury, which together exhibit significantly synergistic and neurotoxic effects [62,63]; **(ii)** aluminum’s effects on nucleoprotein modifications including the epigenetic modification of histones and other DNA-associated nucleoproteins [64]; **(iii)** the association and/or interaction of aluminum-modified and/or misfolded proteins with the pathological hallmarks of neurodegenerative disease that include prions, Aβ peptides, tau proteins and α-synuclein [34,35,36,37,38,65]; **(iv)** the neutralization of aluminum neurotoxicity and genotoxicity including multiple chelation approaches and the use of both synthetic and naturally occurring compounds [66,67,68,69]; and **(v)** the potential for the interaction of aluminum with other oxygen-donor ligands which occur at relatively high densities within the natural cellular and nuclear compartments of all eukaryotic organisms.

## Figures and Tables

**Figure 1 molecules-27-05123-f001:**
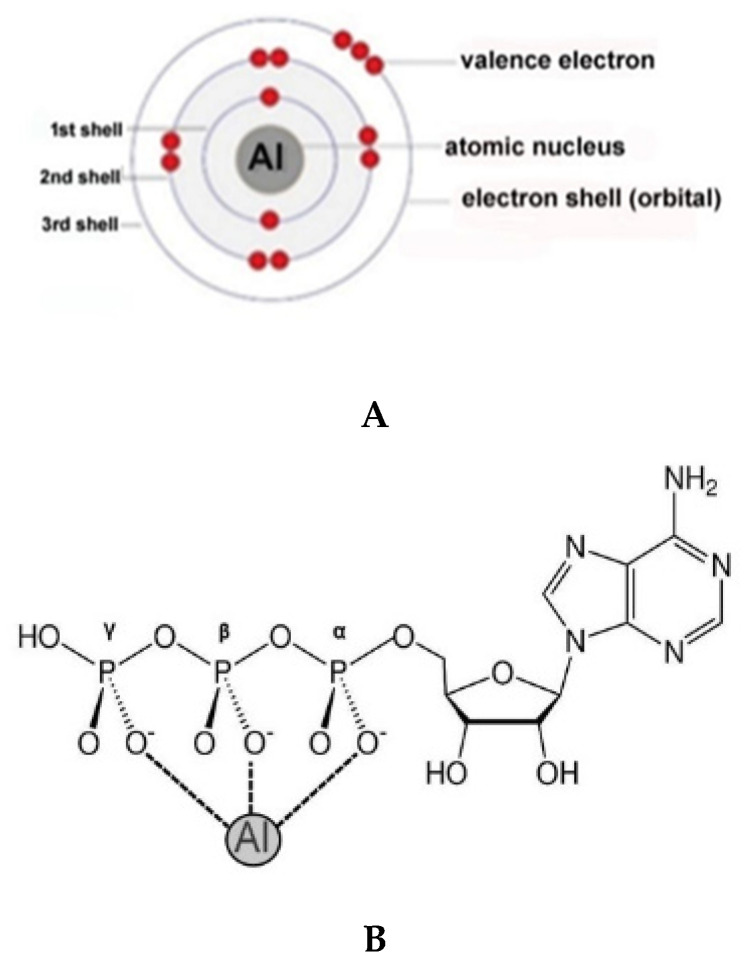
Aluminum (Al^3+^) and adenosine triphosphate (ATP): (**A**) atomic and electronic structure of aluminum; distribution of electrons (red spheres) in aluminum [Al^3+^; Al(III); atomic mass 26.9815; neutral atom electron configuration: 1s^2^2s^2^2p^6^3s^2^3p^1^]; (**B**) potential interaction of highly electropositive trivalent aluminum [Al^3+^; Al(III)] with the α-, β-, and γ-phosphate of adenosine triphosphate (ATP); this interaction strongly stabilizes ATP making it unusable for other biological reactions and/or functions; Al^3+^ undergoes ligand exchange reactions much more slowly than most metal ions and about ~10^5^ times slower than Mg^2+^ [3,4,5]; other Al-ATP, aluminum–adenosine–diphosphate (Al-ADP) and/or aluminum–adenosine–monophosphate (Al-AMP) and/or other coordination structures may be possible under defined physiological situations [3,4,5,6,7,8,23,24,25,26].

**Figure 2 molecules-27-05123-f002:**
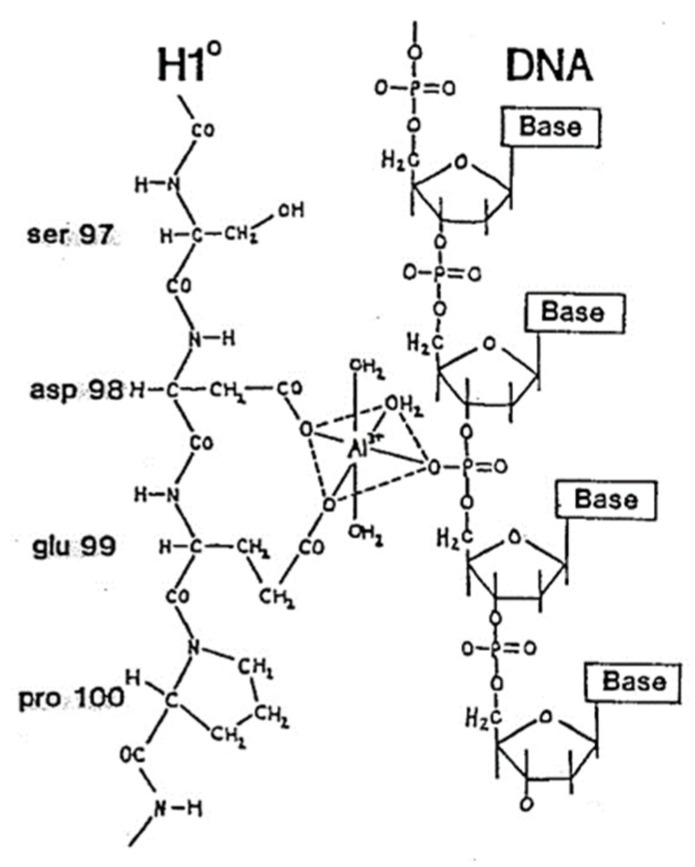
Al^3+^, DNA and linker histone H1°- Proposed 3-dimensional (3D) model for increased stability of H1°–DNA interaction in the presence of hydroxylated Al^3+^- hypothetical interaction of asp98 and glu-99 (D98-E99) of the ~21.4 kDa, ~194 amino acid human H1°, aluminum and high probability target DNA in the 5′ region of the single copy human NF-L promoter; human brain specific H1 linker histones provide unique sites for protein amino acid–aluminum–DNA coordination; such structures may be responsible for the observed increase in binding of linker histones in aluminum-treated neocortical nuclei or in AD brain; such structures would be expected to increase the affinity of linker histones for DNA and increase and stabilize chromatin compaction. One consequence of the enhanced stability of deoxyribonucleoprotein complexes in AD affected neocortical nuclei appears to be a shift to higher order chromatin structure and an ensuing reduction in the transcription of brain-specific genetic information [16,17,18,31,32,33,34]; figure adapted from Figure 30 in reference [32].

**Figure 3 molecules-27-05123-f003:**
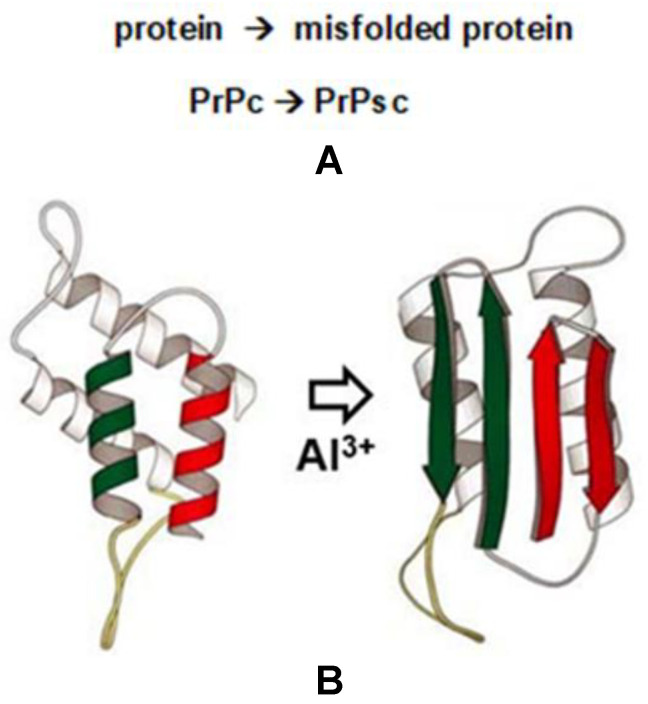
Al^3+^ and the prion PrPc to PrPsc transition in neurodegenerative disease-prion disease (PrD) in mammals appears to be caused by a conformational transition from the cellular prion protein’s native conformation (PrPc) into a pathological isoform called “prion protein-scrapie” (PrPsc); multiple prion-associated neurodegenerative disorders are a consequence of protein misfolding, aggregation, and spread; (**A**) graphical representation of the scheme of the structural transition of the prion protein-cellular (PrPc) native form to the prion-scrapie isoform PrPsc; (**B**) model of the structure of the α-helical-enriched cellular prion protein (PrPc; red and green alpha-helices) to the pathological (abnormal) β-pleated sheet-enriched prion protein (PrPsc; red and green anti-parallel arrows); published evidence indicates that trivalent aluminum (Al**^3+^**; Al(III)) exacerbates both amyloid formation into insoluble aggregates from naturally-occurring Aβ peptides and increases the rate of the onset of AD-type symptomology in transgenic murine models of AD (TgAD); AD is a complex neurological disorder and unique in that it may represent a ‘double prion’ disorder involving both aggregated tau proteins (as a tauopathy) and Aβ peptides (as an amyloidopathy) [34,35,36,37,38]; preliminary monoclonal PrPsc antibody-based evidence further suggests that trivalent aluminum (Al**^3^**^+^; Al(III)) also promotes the misfolding of PrPc into the PrPsc isoform (see manuscript text); structures adapted from references [3,34].

## Data Availability

All data in this Communication have been derived from the quoted References freely- and openly accessible at the National Institutes of Health (NIH) National Library of Medicine at PubMed.gov (https://pubmed.ncbi.nlm.nih.gov; last accessed 13 June 2022).

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
