# Peer review of "Alteration of Biomolecular Conformation by Aluminum-Implications for Protein Misfolding Disease"

_molecules, 2022, doi:10.3390/molecules27165123_

Round 1

Reviewer 1 Report

I suggest to accept the article after minor revisions

Reviewer 2 Report

The authors base their communication on the premise that aluminum has unique biochemical and biophysical properties that make it highly detrimental to the structural integrity, reactivity, and stability of various biomolecules. These include: (i) small size and highly electrophilic nature, (ii) form stable electrostatic bonds, (iii) ability to interact with biomolecules irreversibly, (iv) extremely dense electropositive charge, and (v) most abundant metal in the biosphere. They use three examples to demonstrate that aluminum interacts with biomolecules, which are crucial for energy metabolism, gene expression, cell homeostasis, and signaling pathways. Based on available data, the authors speculate that metals such as aluminum may induce protein misfolding and thus promote the development of neurodegenerative diseases in humans.

Although their findings contribute to the field, the authors must address the following concerns properly before publishing the communication.

The following are examples of the stated above.

1. Abstract: Unnecessarily long. Please define the study question/hypothesis and emphasize the prose to results/findings and their meaning (in simple words).

2. Introduction: Although it contains the scientific foundations that support study, the review of the field's current state is extensive and sometimes redundant, losing the communication's aim. On the other hand, some statements fail meaning due to apparent drafting errors (this applies to the entire manuscript). Hence, reduce the content to important data supporting the study's uniqueness and review the writing style (i.e., reader-based prose).

3. In the second part of the main text, I am confused: the authors propose their manuscript as a short communication, but to me it looks like a mini review. No methods were identified to assess its impact as a scientific communication and, on the other hand, the review is forced to fulfill a range of agent-impact associations in cell biology.

4. Considering the above, I recommend that they must conduct a thorough review of current knowledge about the field, focus on a single agent-impact association, and expand their discussion with a model. Finally, a thorough English language review is also recommended.

Therefore, the current manuscript version is not endorsed for publication.

Reviewer 3 Report

Major revision recommended

Round 2

Reviewer 3 Report

I accept the paper in this form.

Please change the type of the paper from communication to minireview.

Author Response

1 August 2022 

Dear Ricky Guo 

Dear Academic Editor and Reviewer(s)  

Re: our ‘Molecules’ article entitled ‘Alteration of biomolecular conformation by aluminum – implications in protein misfolding disease’ 

As you have suggested the following corrections-changes-upgrades and clarifications have been made in our revised manuscript text in response to the queries and comments raised: 

[1] The accuracy of all names and affiliations have been checked for accuracy and they are all OK as is; 

[2] The manuscript will appear as a ‘Mini-Review’ and any reference to a ‘Communication’ throughout the manuscript text has been removed; 

[3] As requested by Editor/Reviewers a clarified ‘Funding’ statement, ‘Institutional Review Board and Ethics Statement’ and ‘Data Availability Statement’ have been added and/or clarified in the revised manuscript text; 

[4] In the ‘authors contributions’ reference to ‘authors performing experiments’ has been removed and the note on ‘Authors Contributions’ has been changed to: 

Author Contributions: YZ, AIP, PNA, LGB, WL, VRJ and WJL collected, analyzed, distilled and summarized the current literature from the 70 references listed; WJL (communicating author) wrote the article. 

Please let us know that you have received these corrections and revised manuscript;

Also, please let us know if you require anything else 

Thank you kindly 

======================================  

Walter J. Lukiw BS, MS, PhD, Professor of Neurology, Neuroscience and Ophthalmology, Bollinger Professor of Alzheimer's disease (AD), LSU Neuroscience Center, Louisiana State University Health Sciences Center, 2020 Gravier Street, Room 904, New Orleans LA 70112-2272 USA, TEL +1-504-599-0842; EMAIL [email protected]